# Handling Cold-Start Problem in Review Spam Detection by Jointly Embedding Texts and Behaviors

## Abstract

Solving cold-start problem in review spam detection is an urgent and significant task. It can help the on-line review websites to relieve the damage of spammers in time, but has never been investigated by previous work. This paper proposes a novel neural network model to detect review spam for cold-start problem, by learning to represent the new reviewers' review with jointly embedded textual and behavioral information. Experimental results prove the proposed model achieves an effective performance and possesses preferable domain-adaptability. It is also applicable to a large scale dataset in an unsupervised way.

## 1 Introduction

With the rapid growth of products reviews at the web, it has become common for people to read reviews before making purchase decision. The reviews usually contain abundant consumers' personal experiences. It has led to a significant influence on financial gains and fame for businesses. Existing studies have shown that an extra half-star rating on Yelp causes restaurants to sell out 19% points more frequently (Anderson and Magruder, 2012), and a one-star increase in Yelp rating leads to a 5-9 % increase in revenue (Luca, 2011). This, unfortunately, gives strong incentives for imposters (called spammers) to game the system. They post fake reviews or opinions (called review spam) to promote or to discredit some targeted products and services. The news from BBC has shown that around 25% of Yelp reviews could be fake.[1] Therefore, it is urgent to detect review spam, to ensure that the online review continues to be trusted.

Jindal and Liu (2008) make the first step to detect review spam. Most efforts are devoted to explore effective linguistic and behavioral features by subsequent work to distinguish such spam from the real reviews. However, to notice such patterns or form behavioral features, developers should take long time to observe the data, because the features are based on statistics. For instance, the feature *activity window* proposed by Mukherjee et al. (2013c) is to measure the activity freshness of reviewers. It usually takes several months to count the difference of timestamps between the last and first reviews for reviewers. When the features show themselves finally, some major damages might have already been done. Thus, *it is important to design algorithms that can detect review spam as soon as possible, ideally, right after they are posted by the new reviewers*. It is a cold-start problem which is the focus of this paper.

In this paper, we assume that we must identify fake reviews immediately when a new reviewer posts just one review. Unfortunately, it is very difficult because the available information for detecting fake reviews is very poor. Traditional behavioral features based on the statistics can only work well on users' abundant behaviors. The more behavioral information obtained, the more effective the traditional behavioral features are (see experiments in Section 3 ). In the scenario of cold-start, a new reviewer only has a behavior: post a review. As a result, we can not get effective behavioral features from the data. Although, the linguistic features of reviews do not need to take much time to form, Mukherjee et al. (2013c) have proved that the linguistic features are not effective enough in detecting real-life fake reviews from the commercial websites, where we also obtain the same observation (the details are shown in Section 3).

---

[1] http://www.bbc.com/news/technology-24299742

Therefore, the main difficulty of the cold-start spam problem is that there is no sufficient behaviors of the new reviewers for constructing effective behavioral features. Nevertheless, there are ample textual and behavioral information contained in the abundant reviews posted by the existing reviewers (Figure 1). We could employ behavioral information of existing similar reviewers to a new reviewer to approximate his behavioral features. We argue that a reviewer's individual characteristics such as background information, motivation and interactive behavior style have a great influence on a reviewer's textual and behavioral information. So the textual information and the behavioral information of a reviewer are correlated with each other (similar argument in Li et al. (2016)). For example, the students of college are likely to choose the youth hostel during summer vacation, and tend to comment the room price in their reviews. But the financial analysts on business trip may tend to choose the business hotel, the environment and service are what they care about in their reviews.

To augment the behavioral information of the new reviewers in the cold-start problem, we first try to find the textual information which is similar with that of the new reviewer, from the existing reviews. There are several ways to model the textual information of the review spam, such as Unigram (Mukherjee et al., 2013c), POS (Ott et al., 2011) and LIWC (Linguistic Inquiry and Word Count) (Newman et al., 2003). We employ the CNN (Convolutional Neural Network) to model the review text, which has been proved that it can capture complex global semantic information that is difficult to express using traditional discrete manual features (Ren and Zhang, 2016). Then we employ the behavioral information which is correlated with the found textual information to approximate the behavioral information of the new reviewer. An intuitive approach is to search the most similar existing review for the new review, then take the found reviewer's behavioral features as the new reviewers' features (detailed in Section 5.3). However, there are abundant behavioral information in the review graph (Figure 1), it is difficult for the traditional discrete manual behavioral features to record the global behavioral information (Wang et al., 2016). Moreover, the traditional features can not capture the reviewer's individual characteristics, because there is no explicit characteristic tag available in the review system (experi-

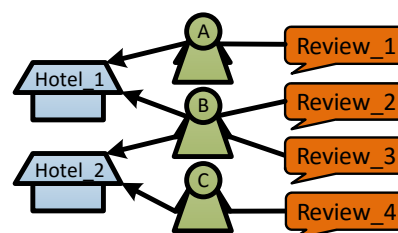

Figure 1: Part of review graph simplified from Yelp.

ments in Section 5.3). So, we propose a neural network model to jointly encode the textual and behavioral information into the review embeddings for detecting the review spam in cold-start problem. By encoding the review graph structure (Figure 1), the proposed model can record the global footprints of the existing reviewers in an unsupervised way, and further record the reviewers' latent characteristic information in the footprints. The jointly learnt review embeddings can model the correlation of the reviewers' textual and behavioral information. When a new reviewer posts a review, the proposed model can represent the review with the similar textual information and the correlated behavioral information encoded in the word embeddings. Finally, the embeddings of the new review are fed into a classifier to identify whether it is spam or not.

In summary, our major contributions include:

- To our best knowledge, this is the first work that explores the cold-start problem in review spam detection. We qualitatively and quantitatively prove that the traditional linguistic and behavioral features are not effective enough in detecting review spam for the cold-start task.

- We propose a neural network model to jointly encode the textual and behavioral information into the review embeddings for the cold-start spam detection task. It is an unsupervised distributional representation model which can learn from large scale unlabeled review data.

- Experimental results on two domains (hotel and restaurant ) give good confidence that the proposed model performs effectively in the cold-start spam detection task.

## 2 Related Work

Jindal and Liu (2008) make the first step to detect review spam. Subsequent work devoted most efforts to explore effective features and spammer-like clues.

**Linguistic features:** Ott et al. (2011) applied psychological and linguistic clues to identify review spam; Harris (2012) explored several writing style features. Syntactic stylometry for review spam detection was investigated in Feng et al. (2012a); Xu and Zhao (2012) using deep linguistic features for finding deceptive opinion spam; Li et al. (2013) studied the topics in the review spam; Li et al. (2014b) further analyzed the general difference of language usage. Fornaciari and Poesio (2014) proved the effectiveness of the N-grams in detecting deceptive Amazon book reviews. The effectiveness of the N-grams was also explored in Cagnina and Rosso (2015). Li et al. (2014a) proposed a positive-unlabeled learning method based on unigrams and bigrams; Kim et al. (2015) carried out a frame-based deep semantic analysis. Hai et al. (2016) exploited the relatedness of multiple review spam detection tasks and available unlabeled data to address the scarcity of labeled opinion spam data by using linguistic features. Besides, (Ren and Zhang, 2016) proved that the CNN model is more effective than the RNN and the traditional discrete manual linguistic features. Hovy (2016) used N-gram generative models to produce reviews and evaluated their effectiveness.

**Behavioral features:** Lim et al. (2010) analyzed reviewers' rating behavioral features; Jindal et al. (2010) identified unusual review patterns which can represent suspicious behaviors of reviews; Li et al. (2011) proposed a two-view semi-supervised co-training method base on behavioral features. Feng et al. (2012b) study the distributions of individual spammers' behaviors. The group spammers' behavioral features were studied in Mukherjee et al. (2012). Temporal patterns of spammers were investigated by Xie et al. (2012), Fei et al. (2013); Li et al. (2015) explored the temporal and spatial patterns. The review graph was analyzed by Wang et al. (2011), Akoglu et al. (2013); Mukherjee et al. (2013a) studied the spamicity of reviewers. Mukherjee et al. (2013c), Mukherjee et al. (2013b) proved that reviewers' behavioral features are more effective than reviews' linguistic features for detecting review spam. Based on this conclusion, recently, researchers (Rayana and Akoglu, 2015; KC and Mukherjee, 2016) have put more efforts in employing reviewers' behavioral features for detecting review spam, the intuition behind which is to capture the reviewers' actions and supposes that

| Features | P | R | F1 | A |
|---|---|---|---|---|
| LF | 54.5 | 71.1 | 61.7 | 55.9 |
| LF+BF | 63.4 | 52.6 | 57.5 | 61.1 |
| LF+BF_abundant | 69.1 | 63.5 | 66.2 | 67.5 |

(a) Hotel

| Features | P | R | F1 | A |
|---|---|---|---|---|
| LF | 53.8 | 80.8 | 64.6 | 55.8 |
| LF+BF | 58.1 | 61.2 | 59.6 | 58.5 |
| LF+BF_abundant | 56.6 | 78.2 | 65.7 | 59.1 |

(b) Restaurant

Table 1: SVM classification results across linguistic features (LF, bigrams here (Mukherjee et al., 2013b)), behavioral features (BF: RL, RD, M-CS (Mukherjee et al., 2013b)) and behavioral features with abundant behavioral information (BF_abundant). Both training and testing use balanced data (50:50).

those reviews written with spammer-like behaviors would be spam. Wang et al. (2016) explored a method to learn the review representation with global behavioral information.

## 3 Whether Traditional Features are Effective

As a new reviewer posted just one review and we have to identify it immediately, the major challenge of the cold-start task is that, the available informations about the new reviewer are very poor. The new reviewer only provide us with one review record. For most traditional features based on the statistics, they can not form themselves or make no sense, such as the *percentage of reviews written at weekends* (Li et al., 2015), the *entropy of rating distribution of user's review* (Rayana and Akoglu, 2015). To investigate whether traditional features are effective in the cold-start task, we conducted experiments on the Yelp dataset in Mukherjee et al. (2013c). We trained SVM models with different features on the existing reviews posted before January 1, 2012, and tested on the new reviews which just posted by the new reviewers after January 1, 2012. Results are shown in Table 1.

### 3.1 Linguistic Features' Poor Performance

The linguistic features need not to take much time to form. But Mukherjee et al. (2013c) have proved that the linguistic features are not effective enough in detecting real-life fake reviews from the commercial websites, compared with the performances on the crowd source datasets (Ott et al.,

2011). They showed that the word bigrams perform better than the other linguistic features, such as LIWC (Newman et al., 2003; Pennebaker et al., 2007), part-of-speech sequence patterns (Mukherjee and Liu, 2010), deep syntax (Feng et al., 2012a), information gain (Mukherjee et al., 2013c) and so on. So, we conduct experiments with the word bigrams feature. As shown in Table 1 (a, b) row 1, the word bigrams result in only around 55% in accuracy in both the hotel and restaurant domains. It indicates that the most effective traditional linguistic feature (i.e., the word bigrams) can't detect the review spam effectively in the cold start task.

### 3.2 Behavioral Features only Work Well with Abundant Information

Because there is not enough available information about the new reviewer, for most traditional behavioral features based on the statistical mechanism, they couldn't form themselves or make no sense. We investigated the previous work and found that there are three behavioral features can be applied to the cold-start task. They are proposed by Mukherjee et al. (2013b), i.e., 1.*Review length (RL)* : the length of the new review posted by the new reviewer; 2.*Reviewer deviation (RD)*: the absolute rating deviation of the new reviewer's review from other reviews on the same business; 3.*Maximum content similarity (MCS)* : the maximum content similarity (using cosine similarity) between the new reviewer's review with other reviews on the same business.

Table 1 (a, b) row 2 shows the experiment results by the combinations of the bigrams feature and the three behavioral features described above. The behavioral features make around 5% improvement in accuracy in the hotel domain (2.7% in the restaurant domain) as compared with only using bigrams. The accuracy is improved but it is just near 60% in average. It indicates that the traditional features are not effective enough with poor behavioral information. What's more, the behavioral features cause around 4.6% decrease in F1-score and around 19% decrease in Recall in both hotel and restaurant domains. It is obvious that there is more false-positive review spam caused by the behavioral features as compared to only using bigrams. It further indicates that the traditional behavioral features' discrimination for review spam gets to be weakened by the poor behavioral infor-

mation.

To go a step further, we carried experiments with the three behavioral features which are formed on abundant behavioral information. When the new reviewers continue to post more reviews in after weeks, their behavioral information gets to be more. Then the review system could obtain more sufficient data to extract behavior features as compared to the poor information in the cold-start period. So the behavioral features with abundant information make an obvious improvement in accuracy (6.4%) in the hotel domain (Table 1 (a) row 3) as compared with the results in Table 1 (a) row 2. But it is only 0.6% in the restaurant domain. By statistics on the datasets, we found that the new reviewers posted about 54.4 reviews in average after their first post in the hotel domain, but it is only 10 reviews in average for the new reviewers in the restaurant domain. The added behavioral information in the hotel domain is richer than that in the restaurant domain. It indicates that:

- the traditional behavioral features can only work well with abundant behavioral information;
- the more behavioral information can be obtained, the more effective the traditional behavioral features are.

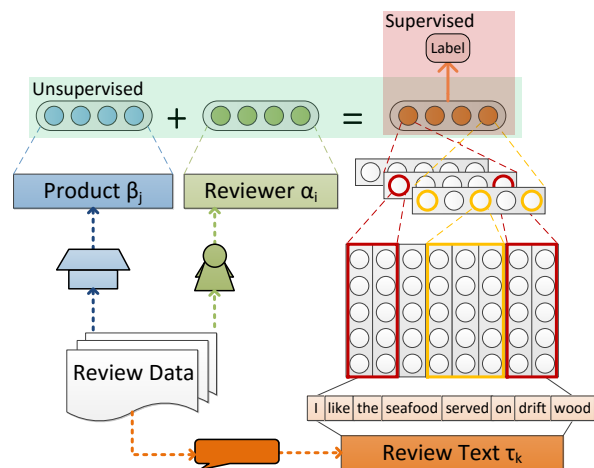

Figure 2: Illustrated of our model.

## 4 The Proposed Model

The difficulty of detecting review spam in the cold-start task is that the available behavioral information of new reviewers is very poor. The new reviewer just posted one review and we have to filter it out immediately, there is not any historical reviews provided to us. As we argued, the textual in-

formation and the behavioral information of a reviewer are correlated with each other. So, to augment the behavioral information of new reviewers, we try to find the textual information which is similar with that of the new reviewer, from existing reviews. Then we take the behavioral information which is correlated with the found textual information as the most possible behavioral information of the new reviewer. For this purpose, we propose a neural network model to jointly encode the textual and behavioral information into the review embeddings for detecting the review spam in the cold-start problem (shown in Figure 2). When a new reviewer posts a review, the neural network can represent the review with the similar textual information and the correlated behavioral information encoded in the word embeddings. Finally, embeddings of the new review are fed into a classifier to identify whether it is spam or not.

### 4.1 Behavioral Information Encoding

In Figure 1, there is a part of review graph which is simplified from the Yelp website. As it shows, the review graph contains the global behavioral information (footprints) of the existing reviewers. Because the motivations of the spammers and the real reviewers are totally different, the distributions of the behavioral information of them are different (Mukherjee et al., 2013a). There are businesses (even highly reputable ones) paying people to write fake reviews for them to promote their products/services and/or to discredit their competitors (Liu, 2015). So the behavioral footprints of the spammers are decided by the demands of the businesses. But the real reviewers only post reviews to the product or services they have actually experienced. Their behavioral footprints are influenced by their own characteristics. Previous work extracts behavioral features for reviewers from these behavioral information. But it is impractical to the new reviewers in the cold-start task. Moreover, the traditional discrete features can not effectively record the global behavioral information (Wang et al., 2016). Besides, there is no explicit characteristic tag available in the review system, and we need to find a way to record the reviewers' latent characters information in footprints.

Therefore we encode these behavioral information into our model by utilizing a embedding learning model which is similar with TransE (Bordes et al., 2013). TransE is a model which can encode the graph structure, and represent the nodes

and edges (head, translation/relation, tail) in low dimension vector space. TransE has been proved that it is good at describing the global information of the graph structure by the work about distributional representation for knowledge base (Guu et al., 2015). We consider that each reviewer in review graph describes the product in his/her own view and writes the review. When we represent the product, reviewer and review in low dimension vector space, the reviewer embeddings can be taken as a translation vector, which has translated the product embeddings to the review embeddings. So, as shown in Figure 2, we take the products (hotels/restaurants) as the head part of the TransE network in our model, take the reviewers as the translation (relation) part and take the review as the tail part. By learning from the existing large scale unlabeled reviews of the review graph, we can encode the global behavioral information into our model without extracting any traditional behavioral feature, and record reviewers' latent characteristics information.

More formally, we minimize a margin-based criterion over the training set:

$$\mathcal{L} = \sum_{(\boldsymbol{\beta},\boldsymbol{\alpha},\boldsymbol{\tau}) \in S} \sum_{(\boldsymbol{\beta}',\boldsymbol{\alpha},\boldsymbol{\tau}') \in S'} \max \quad (1)$$
$$\{0, 1 + d(\boldsymbol{\beta} + \boldsymbol{\alpha}, \boldsymbol{\tau}) - d(\boldsymbol{\beta}' + \boldsymbol{\alpha}, \boldsymbol{\tau}')\}$$

$S$ denotes the training set of triples $(\boldsymbol{\beta}, \boldsymbol{\alpha}, \boldsymbol{\tau})$ composed product $\boldsymbol{\beta}$ ($\boldsymbol{\beta} \in B$, products set (head part)), reviewer $\boldsymbol{\alpha}$ ($\boldsymbol{\alpha} \in A$, reviewers set (translation part)) and review text embeddings learnt by the CNN $\boldsymbol{\tau}$ ($\boldsymbol{\tau} \in T$, review texts set (tail part)).

$$S' = \{(\boldsymbol{\beta}',\boldsymbol{\alpha},\boldsymbol{\tau})|\boldsymbol{\beta}' \in B\} \cup \{(\boldsymbol{\beta},\boldsymbol{\alpha},\boldsymbol{\tau}')|\boldsymbol{\tau}' \in T\} \quad (2)$$

The set of corrupted triplets $S'$ (Equation (2)), is composed of training triplets with either the product or review text replaced by a random chosen one (but not both at the same time).

$$d(\boldsymbol{\beta} + \boldsymbol{\alpha}, \boldsymbol{\tau}) = \|\boldsymbol{\beta} + \boldsymbol{\alpha} - \boldsymbol{\tau}\|_2^2,$$
$$s.t. \|\boldsymbol{\beta}\|_2^2 = \|\boldsymbol{\alpha}\|_2^2 = \|\boldsymbol{\tau}\|_2^2 = 1 \quad (3)$$

$d(\boldsymbol{\beta} + \boldsymbol{\alpha}, \boldsymbol{\tau})$ is the dissimilarity function with the squared euclidean distance.

### 4.2 Textual Information Encoding

To encode the textual information into our model, we adopt a convolutional neural network (CNN) to learn to represent the existing reviews. By statistics, we find that a review usually refers to several aspects of the products or services. For example, a hotel review may comment the room price, the free

WiFi and the bathroom at the same time. Compared with the recurrent neural network (RNN), the CNN can do a better job of modeling the different aspects of a review. Ren and Zhang (2016) have proved that the CNN can capture complex global semantic information and detect review spam more effectively, compared with traditional discrete manual features and the RNN model. As shown in Figure 2, we take the learnt embeddings $\tau$ of reviews by the CNN as the tail part.

Specifically, we denote the review text consisting of $n$ words as $\{w_1, w_2, ..., w_n\}$, the word embeddings $e(w_i) \in R^D$, $D$ is the word vector dimension. We take the concatenation of the word embeddings in a fixed length window size $Z$ as the input of the linear layer, which is denoted as $I_i \in R^{D \times Z}$. So the output of the linear layer $H_i$ is calculated by $H_{k,i} = W_k \cdot I_i + b_i$, where $W_k \in R^{D \times Z}$ is the weight matrix of filter $k$. We utilize a max pooling layer to get the output of each filter. Then we take $tanh$ as the activation function and concatenate the outputs as the final review embeddings, which is denoted as $\tau_i$.

### 4.3 Jointly Information Encoding

To model the correlation of the textual and behavioral information, we employ the jointly information encoding. By jointly learning from the global review graph, the textual and behavioral information of existing spammers and real reviewers are embedded into the word embeddings.

In addition, the rating usually represents the sentiment polarity of a review, e.g., five star means 'like' and one star means 'dislike'. The spammers often review their target products with low rating for discredited purpose, and with high rating for promoted purpose. To encode the semantics of the sentiment polarity into the review embeddings, we learn the embeddings of 1-5 stars rating in our model at the same time. They are taken as the constraints of the review embeddings during the joint learning. They are calculated as:

$$\mathcal{C} = \sum_{(\boldsymbol{\tau},\boldsymbol{\gamma}) \in \Gamma} \sum_{(\boldsymbol{\tau},\boldsymbol{\gamma}') \in \Gamma'} \max\{0, 1 + g(\boldsymbol{\tau},\boldsymbol{\gamma}) - g(\boldsymbol{\tau},\boldsymbol{\gamma}')\} \quad (4)$$

The set of corrupted tuples $\Gamma'$ is composed of training tuples $\Gamma$ with the rating of review replaced by its opposite rating (i.e., 1 by 5, 2 by 4, 3 by 1 or 5). $g(\boldsymbol{\tau},\boldsymbol{\gamma}) = \|\boldsymbol{\tau} - \boldsymbol{\gamma}\|_2^2$, norm constraints: $\|\boldsymbol{\gamma}\|_2^2 = 1$.

The final joint loss function is as follows:

$$\mathcal{L}_{\mathcal{J}} = (1 - \theta)\mathcal{L} + \theta\mathcal{C} \quad (5)$$

where $\theta$ is a hyper-parameter.

| Domain | Hotel | Restaurant |
|---|---|---|
| #reviews | 688328 | 788471 |
| date range | 2004.10.23 2012.09.26 | 2004.10.12 2012.10.02 |
| %before 2012.01.01 | 99.01% | 97.40% |

Table 2: Yelp Whole Dataset Statistics (Labeled and Unlabeled).

| Domain | Hotel | Restaurant |
|---|---|---|
| fake | 802 | 8368 |
| non-fake | 4876 | 50149 |
| %fake | 14.1% | 14.3% |
| #reviews | 5678 | 58517 |
| #reviewers | 5124 | 35593 |

Table 3: Yelp Labeled Dataset Statistics.

## 5 Experiments

### 5.1 Datasets and Evaluation Metrics

**Datasets:** To evaluate the proposed method, we conducted experiments on Yelp dataset that was used in (Mukherjee et al., 2013b,c; Rayana and Akoglu, 2015). The statistics of the Yelp dataset are listed in Table 2 and Table 3. The reviewed product here refers to a hotel or restaurant. We take the existing reviews posted before January 1, 2012 as the training datasets, and take the first new reviews which just posted by the new reviewers after January 1, 2012 as the test datasets.

**Evaluation Metrics:** We select precision (P), recall (R), F1-Score (F1), accuracy (A) as metrics.

### 5.2 Our Model v.s. the Traditional Features

To illustrate the effectiveness of our model, we conduct experiments on the public datasets, and make comparison with the most effective traditional linguistic features, e.g., bigrams, and the three practicable traditional behavioral features (RL, RD, MCS (Mukherjee et al., 2013b)) referred in Section 3.2. The results are shown in Table 4. For our model, we set the dimension of embeddings to 100, the number of CNN filters to 100, $\theta$ to 0.1, $Z$ to 2. The hyper-parameters are tuned by grid search on the development dataset. The product and reviewer embeddings are randomly initialized from a uniform distribution (Socher et al., 2013). The word embeddings are initialized with 100-dimensions vectors pre-trained by the CBOW model (Word2Vec) (Mikolov et al., 2013).As Table 4 showed, our model observably performs better in detecting review spam for the cold-start task in both hotel and restaurant domains.

| Features | P | R | F1 | A | | P | R | F1 | A | |
|---|---|---|---|---|---|---|---|---|---|---|
| LF | 54.5 | 71.1 | 61.7 | 55.9 | 1 | 53.8 | 80.8 | 64.6 | 55.8 | 1 |
| LF+BF | 63.4 | 52.6 | 57.5 | 61.1 | 2 | 58.1 | 61.2 | 59.6 | 58.5 | 2 |
| BF_EditSim+LF | 55.3 | 69.7 | 61.6 | 56.6 | 3 | 53.9 | 82.2 | 65.1 | 56.0 | 3 |
| BF_W2Vsim+W2V | 58.4 | 65.9 | 61.9 | 59.5 | 4 | 56.3 | 73.4 | 63.7 | 58.2 | 4 |
| Ours_RE | 62.1 | 68.3 | **65.1** | **63.3** | 5 | 58.4 | 75.1 | **65.7** | **60.8** | 5 |
| Ours_RE+RRE+PRE | 63.6 | 71.2 | **67.2** | **65.4** | 6 | 59.0 | 78.8 | **67.5** | **62.0** | 6 |

(a) Hotel                                      (b) Restaurant

Table 4: SVM classification results across linguistic features (LF, bigrams here (Mukherjee et al., 2013b)), behavioral features (BF: RL, RD, MCS (Mukherjee et al., 2013b)); the SVM classification results by the intuitive method that finding the most similar existing review by edit distance ratio and take the found reviewers' behavioral features as approximation (BF_EditSim+LF), and results by the intuitive method that finding the most similar existing review by averaged pre-trained word embeddings (using Word2Vec) (BF_W2Vsim+W2V); and the SVM classification results across the learnt review embeddings (RE), the learnt review's rating embeddings (RRE), the learnt product's average rating embeddings (PRE) by our model. Improvements of our model are statistically significant with p<0.005 based on paired *t*-test.

**Review Embeddings** Compared with the traditional linguistic features, e.g., bigrams, using the review embeddings learnt by our model, results in around 3.4% improvement in F1 and around 7.4% improvement in A in the hotel domain (1.1% in F1 and 5.0% in A for the restaurant domain, shown in Tabel 4 (a,b) rows 1, 5). Compared with the combination of the bigrams and the traditional behavioral features, using the review embeddings learnt by our model, results in around 7.6% improvement in F1 and around 2.2% improvement in A in the hotel domain (6.1% in F1 and 2.3% in A for the restaurant domain, shown in Tabel 4 (a,b) rows 2, 5). The F1-Score (F1) of the classification under the balance distribution reflects the ability of detecting the review spam. The accuracy (A) of the classification under the balance distribution reflects the ability of identifying both the review spam and the real review. The experiment results indicate that our model performs significantly better than the traditional methods in F1 and A at the same time. The learnt review embeddings with encoded linguistic and behavioral information are more effective in detecting review spam for the cold-start task.

**Rating Embeddings** As we referred in Section 4.3, the rating of a review usually means the sentiment polarity of a real reviewer or the motivation of a spammer. As shown in Table 4 (a,b) rows 6, adding the rating embeddings of the products (hotel/restaurant) and reviews renders even higher F1 and A. We suppose that different rating embeddings are encoded with different semantic meanings. They reflect the semantic divergences be-

tween the average rating of the product and the review rating. In results, using RE+RRE+PRE which makes the best performance of our model, results in around 5.5% improvement in F1 and around 9.5% improvement in A in the hotel domain (2.9% in F1 and 6.2% in A for the restaurant domain, shown in Tabel 4 (a,b) rows 1, 6), compared with the LF. Using RE+RRE+PRE results in around 9.7% improvement in F1 and around 4.3% improvement in A in the hotel domain (7.9% in F1 and 3.5% in A for the restaurant domain, shown in Tabel 4 (a,b) rows 2, 6), compared with the LF+BF.

The experiment results proves that our model is effective. The improvements in both the F1 and A prove that our model performs well in both detecting the review spam and identifying the real review. Furthermore, the improvements in both the hotel and restaurant domains prove that our model possesses preferable domain-adaptability [2]. It can learn to represent the reviews with global linguistic and behavioral information from large scale unlabeled existing reviews.

### 5.3 Our Jointly Embeddings v.s. the Intuitive Methods

As mentioned in Section 1, to approximate the behavioral information of the new reviewers, there are other intuitive methods. So we conduct experiments with two intuitive methods as com-

---

[2]The improvements in hotel domain are greater than that in restaurant domain. The possible reason is the proportion of the available training data in hotel domain is higher than that in restaurant domain (99.01% vs. 97.40% in Table 2).

| Features | P | R | F1 | A | | P | R | F1 | A | |
|---|---|---|---|---|---|---|---|---|---|---|
| LF | 54.5 | 71.1 | 61.7 | 55.9 | 1 | 53.8 | 80.8 | 64.6 | 55.8 | 1 |
| Ours_CNN | 61.2 | 51.7 | 56.1 | 59.5 | 2 | 56.9 | 58.8 | 57.8 | 57.1 | 2 |
| Ours_RE | 62.1 | 68.3 | **65.1** | **63.3** | 3 | 58.4 | 75.1 | **65.7** | **60.8** | 3 |

(a) Hotel (b) Restaurant

Table 5: SVM classification results across linguistic features (LF, bigrams here (Mukherjee et al., 2013b)), the learnt review embeddings (RE) ; and the classification results by only using our CNN. Both training and testing use balanced data (50:50). Improvements of our model are statistically significant with p<0.005 based on paired *t*-test.

parison. One is finding the most similar existing review by edit distance ratio and taking the found reviewers' behavioral features as approximation, and then training the classifier on the behavioral features and bigrams (BF_EditSim+LF). The other is finding the most similar existing review by cosine similarity of review embeddings which is the average of the pre-trained word embeddings (using Word2Vec), and then training the classifier on the behavioral features and review embeddings (BF_W2Vsim+W2V). As shown in Table 4, our jointly embeddings (Ours_RE and Ours_RE+RRE+PRE) obviously perform better than the intuitive methods, such as the Ours_RE is 3.8% (Accuracy) and 3.2% (F1) better than BF_W2Vsim+W2V in the hotel domain. The experiments indicate that, our jointly embeddings do a better job in capturing the reviewer's characteristics and modeling the correlation of textual and behavioral information.

### 5.4 The Effectiveness of Encoding the Global Behavioral Information

To further evaluate the effectiveness of encoding the global behavioral information in our model, we build an independent supervised convolutional neural network which has the same structure and parameter settings with the CNN part of our model. There is not any review graphic or behavioral information in this independent supervised CNN (Tabel 5 (a,b) row 2). As shown in Tabel 5 (a,b) rows 2, 3, compared with the review embeddings learnt by the independent supervised CNN, using the review embeddings learnt by our model results in around 9.0% improvement in F1 and around 3.8% improvement in A in the hotel domain (7.9% in F1 and 3.7% in A for the restaurant domain. The results show that our model can represent the new reviews posted by the new reviewers with the correlated behavioral information encoded in the word embeddings. The transE part of our model has effectively recorded the behavioral informa-

tion of the review graph. Thus, our model is more effective by jointly embedding the textual and behavioral informations, it helps to augment the possible behavioral information of the new reviewer.

### 5.5 The Effectiveness of CNN

Compared with the the most effective linguistic features, e.g., bigrams, our independent supervised convolutional neural network performs better in A than F1 (shown in Tabel 4 (a,b) rows 1, 2). It indicates that the CNN do a better job in identifying the real review than the review spam. We suppose that the possible reason is that the CNN is good at modeling the different semantic aspects of a review. And the real reviewers usually tend to describe different aspects of a hotel or restaurant according to their real personal experiences, but the spammers can only forge fake reviews with their own infinite imagination. Mukherjee et al. (2013b) also proved that different psychological states of the minds of the spammers and non-spammers, lead to significant linguistic differences between review spam and non-spam.

## 6 Conclusion and Future Work

This paper analyzes the importance and difficulty of the cold-start challenge in review spam combat. We propose a neural network model that jointly embeds the existing textual and behavioral information for detecting review spam in the cold-start task. It can learn to represent the new review of the new reviewer with the similar textual information and the correlated behavioral information in an unsupervised way. Then, a classifier is applied to detect the review spam. Experimental results prove the proposed model achieves an effective performance and possesses preferable domain-adaptability. It is also applicable to a large scale dataset in an unsupervised way. To our best knowledge, this is the first work to handle the cold-start problem in review spam detection. We are going to explore more effective models in future.

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
