# Peer review of "Handling Cold-Start Problem in Review Spam Detection by Jointly Embedding Texts and Behaviors"

_ACL 2017 — decision unknown_

[Official Review · Reviewer 1 · rating 4 · confidence 4]
soundness 5 · originality 5 · clarity 4 · impact 3 · substance 4 · appropriateness 5 · meaningful comparison 3 · presentation format Oral Presentation

- Strengths:
     - The related work is quite thorough and the comparison with the approach
presented in this paper makes the hypothesis of the paper stronger. The
evaluation section is also extensive and thus, the experiments are convincing.

- Weaknesses:
     - In Section 3 it is not clear what is exactly the dataset that you used
for training the SVM and your own model. Furthermore, you only give the
starting date for collecting the testing data, but there is no other
information related to the size of the dataset or the time frame when the data
was collected. This might also give some insight for the results and statistics
given in Section 3.2.
     - In Table 3 we can see that the number of reviewers is only slightly
lower than the number of reviews posted (at least for hotels), which means that
only a few reviewers posted more than one review, in the labeled dataset. How
does this compare with the full dataset in Table 2? What is the exact number of
reviewers in Table 2 (to know what is the percentage of labeled reviewers)? It
is also interesting to know how many reviews are made by one person on average.
If there are only a few reviewers that post more than one review (i.e., not
that much info to learn from), the results would benefit from a thorough
discussion. 

- General Discussion:
     This paper focuses on identifying spam reviews under the assumption that
we deal with a cold-start problem, i.e., we do not have enough information to
draw a conclusion. The paper proposes a neural network model that learns how to
represent new reviews by jointly using embedded textual information and
behaviour information. Overall, the paper is very well written and the results
are compelling.

- Typos and/or grammar:                                 
     - The new reviewer only provide us                                        

     - Jindal and Liu (2008) make the first step -> the work is quite old, you
could use past tense to refer to it
     - Usage of short form “can’t”, “couldn’t”, “what’s”
instead of the prefered long form
     - The following sentence is not clear and should be rephrased: “The new
reviewer just posted one review and we have to filter it out immediately, there
is not any historical reviews provided to us.“

[Official Review · Reviewer 2 · rating 5 · confidence 5]
soundness 5 · originality 5 · clarity 4 · impact 3 · substance 5 · appropriateness 5 · meaningful comparison 3 · presentation format Oral Presentation

This paper investigates the cold-start problem in review spam detection. The
authors first qualitatively and quantitatively analyze the cold-start problem.
They observe that there is no enough prior data from a new user in this
realistic scenario. The traditional features fail to help to identify review
spam. Instead, they turn to rely on the abundant textual and behavioral
information of the existing reviewer to augment the information of a new user.
In specific, they propose a neural network to represent the review of the new
reviewer with the learnt word embedding and jointly encoded behavioral
information. In the experiments, the authors make comparisons with traditional
methods, and show the effectiveness of their model.

- Strengths:

The paper is well organized and clearly written. The idea of jointly encoding
texts and behaviors is interesting. The cold-start problem is actually an
urgent problem to several online review analysis applications. In my knowledge,
the previous work has not yet attempted to tackle this problem. This paper is
meaningful and presents a reasonable analysis. And the results of the proposed
model can also be available for downstream detection models.

- Weaknesses:

In experiments, the author set the window width of the filters in the CNN
module to 2. Did the author try other window widths, for example width `1' to
extract unigram features, `3' to trigram, or use them together? 
The authors may add more details about the previous work in the related work
section. More specifically description would help the readers to understand the
task clearly.

There are also some typos to be corrected:
Sec 1: ``...making purchase decision...'' should be ``making a/the purchase
decision''
Sec 1: ``...are devoted to explore... '' should be `` are devoted to
exploring''
Sec 1: ``...there is on sufficient behaviors...'' should be “there are no
sufficient behaviors''
Sec 1: ``...on business trip...'' should be ``on a business trip''
Sec 1: ``...there are abundant behavior information...'' should be ``there is
abundant behavior''
Sec 3: ``The new reviewer only provide us...'' should be ``...The new reviewer
only provides us...''
Sec 3: ``...features need not to take much...'' should be ``...features need
not take much...''
Sec 4: ``...there is not any historical reviews...'' should be ``...there are
not any historical reviews...''
Sec 4: ``...utilizing a embedding learning model...'' should be ``...utilizing
an embedding learning model...''
Sec 5.2 ``...The experiment results proves...'' should be ``...The experiment
results prove...''

- General Discussion:

It is a good paper and should be accepted by ACL.